# Multidisciplinary Team Care in Pituitary Tumours

**DOI:** 10.3390/cancers16050950

**Published:** 2024-02-27

**Authors:** Pedro Marques, Amets Sagarribay, Francisco Tortosa, Lia Neto, Joana Tavares Ferreira, João Subtil, Ana Palha, Daniela Dias, Inês Sapinho

**Affiliations:** 1Pituitary Tumor Unit, Endocrinology Department, Hospital CUF Descobertas, 1998-018 Lisbon, Portugaldaniela.dias@cuf.pt (D.D.); ines.santos@cuf.pt (I.S.); 2Faculty of Medicine, Universidade Católica Portuguesa, 1649-023 Lisbon, Portugal; 3Pituitary Tumor Unit, Neurosurgery Department, Hospital CUF Descobertas, 1998-018 Lisbon, Portugal; 4Pituitary Tumor Unit, Pathology Department, Hospital CUF Descobertas, 1998-018 Lisbon, Portugal; francisco.tortosa@cuf.pt; 5Pituitary Tumor Unit, Radiology Department, Hospital CUF Descobertas, 1998-018 Lisbon, Portugal; lia.neto@cuf.pt; 6Pituitary Tumor Unit, Ophthalmology Department, Hospital CUF Descobertas, 1998-018 Lisbon, Portugal; joana.ferreira@cuf.pt; 7Pituitary Tumor Unit, Otorhinolaryngology Department, Hospital CUF Descobertas, 1998-018 Lisbon, Portugal; joao.subtil@cuf.pt

**Keywords:** pituitary tumour, multidisciplinary team care, pituitary tumour centres of excellence (PTCOE)

## Abstract

**Simple Summary:**

Pituitary tumours are slowly growing tumours of the pituitary gland, and they can cause damage due to the invasion or compression of surrounding tissues, such as the nerves responsible for vision, and/or due to altered hormone production. The treatment of pituitary tumours is complex and requires a wide range of medical specialists, including neurosurgeons, endocrinologists, neuropathologists, neuroradiologists, neuro-ophthalmologists, and otorhinolaryngologists, among others. Thus, optimal management of patients with pituitary tumours is best provided in the context of a dedicated multidisciplinary team with expertise in treating such conditions.

**Abstract:**

The optimal care for patients with pituitary tumours is best provided in a multidisciplinary and collaborative environment, which requires the contribution of multiple medical specialties working together. The benefits and advantages of the pituitary multidisciplinary team (MDT) are broad, and all relevant international consensus and guidelines in the field recommend that patients with pituitary tumours should always be managed in a MDT. Endocrinologists and neurosurgeons are normally the leading specialties within the pituitary MDT, supported by many other specialties with significant contributions to the diagnosis and management of pituitary tumours, including neuropathology, neuroradiology, neuro-ophthalmology, and otorhinolaryngology, among others. Here, we review the literature concerning the concepts of Pituitary MDT/Pituitary Tumour Centre of Excellence (PTCOE) in terms of their mission, goals, benefits, structure, proposed models of function, and barriers, and we also provide the views of different specialists involved in our Pituitary MDT.

## 1. Introduction

The concept of multidisciplinary team (MDT) and multidisciplinary medical care is recognised as best practice and has been increasingly used for a wide variety of conditions [1,2]. The contemporaneous management of complex and/or rare disorders relies mainly on MDT meetings and discussions among experienced healthcare professionals, which ensure timely, appropriate, specialized, and multidisciplinary decision-making, thus improving outcomes. Moreover, MDT meetings provide unique opportunities for improved medical communication and development of cohesive management plans, learning platforms, and knowledge for research projects [1,3].

Pituitary tumours arise from the adenohypophysis and account for 10–15% of all intracranial neoplasms. Although these tumours are usually benign, up to 30–45% invade the cavernous or sphenoid sinus [4,5,6]. Less frequently, they may behave aggressively, recur multiple times and/or become resistant to conventional treatments, or more rarely, metastasize [6,7]. From a histological standpoint, pituitary tumours are classified on the basis of the respective cell lineage determined by the expression of pituitary hormones, transcription factors, or other biomarkers as: PIT1-lineage tumours (somatotroph tumours, lactotroph tumours, mammosomatotroph tumours, thyrotroph tumours, mature/immature PIT1 lineage tumours, acidophil stem cell tumours, and mixed somatotroph and lactotroph tumours); TPIT-lineage tumours (corticotroph tumours including densely granulated, sparsely granulated, and Crooke cell tumours); SF1-lineage tumours (i.e., gonadotroph tumours); tumours with no distinct cell lineage (plurihormonal tumours); and null cell tumours [8]. Pituitary tumours may cause syndromes related to excessive hormone secretion, such as Cushing’s disease or acromegaly, or hypopituitarism-related symptoms due to the compression of the normal pituitary. Also, presentation can include neurological symptoms such as visual defects, diplopia, or headache. Moreover, approximately 5% of pituitary tumours may be familial, occurring in isolation or as part of a genetic syndrome such as multiple endocrine neoplasia type 1 or Carney complex [6,9].

The above-mentioned aspects, together with the challenging anatomical location of the pituitary and its relationship with key structures such as the optic chiasm, impose several difficulties on the management of patients with pituitary tumours. Hence, the best care for pituitary tumour patients is provided by an experienced MDT working in an interdisciplinary and collaborative manner within the frame of a Pituitary MDT or Pituitary Tumour Centre of Excellence (PTCOE) [10,11,12].

Here, we reviewed the literature concerning Pituitary MDT and PTCOE, and we are also providing the views and perspectives of different medical specialists involved in Pituitary MDT in our centre.

## 2. Concept and Mission of the Pituitary Multidisciplinary Team (MDT)/Pituitary Tumour Centres of Excellence (PTCOE)

The concept of a Centre of Excellence is disseminating in several fields of medicine to address public and professional concerns regarding the quality of care for patients suffering from a certain condition. This concept is useful for conditions demanding an interdisciplinary management approach, and thus, a dedicated MDT composed by different experts in surgery, medical treatments, radiotherapy, or diagnostic procedures [10,13]. The concepts of MDT, or Centres of Excellence, are relevant in several cancers and rare diseases, as well as in many endocrine conditions, including thyroid diseases [14,15], obesity [16,17,18], diabetes [19,20], and neuroendocrine tumours [21].

The pursuit of excellence in delivering the best care to patients with pituitary tumours and other sellar lesions, including craniopharyngiomas or Rathke’s cleft cysts, has led to the implementation of Pituitary MDT/PTCOE. Pituitary MDT/PTCOE are now widely recommended by different guidelines and medical societies relevant in the field [2,10,11,22,23]. The definition criteria of PTCOE have been proposed [10], and further validated in a survey of several internationally recognised tertiary centres [11]. Organising and delivering multidisciplinary management by experienced pituitary neurosurgeons and endocrinologists, with the support of other key specialists, is critical to the definition of Pituitary MDT/PTCOE [10,11,12].

Management of patients with pituitary diseases has much improved since the implementation of PTCOE, or Pituitary MDT, with a “patient-centric” philosophy where the patient is at the core of its mission, and all activities and goals are aimed at improving the patient’s experience and outcomes [2,10]. The mission and main clinical goals of the Pituitary MDT/PTCOE are summarised in Table 1. Ultimately, the key goal is to eliminate, or at least reduce, the excess morbidity and mortality due to pituitary tumours, pituitary hypersecretion syndromes, and other disorders affecting the pituitary, as well as manage the anterior or posterior pituitary hormone deficiencies [8,10,11,24,25,26,27,28,29,30,31]. For most patients, this requires the establishment of a complex program of care and the need for long-term follow-up [10,11].

## 3. Benefits of the Pituitary MDT/PTCOE

As in other cancers, the dissemination of Pituitary MDT accompanied the improvement of the standards of care and outcomes of pituitary tumour patients over recent years [32,33,34], and it has relied on the notion of being beneficial for patients and healthcare systems, while there has been little evidence supporting their establishment. MDT benefits are partly attributable to better engagement of the relevant medical specialties involved in the management of such patients, while close collaboration prevents complications and facilitates the utilisation of the latest developments, guidelines, and technologies [2,35]. On the other hand, Pituitary MDT for healthcare professionals may be beneficial for professional fulfillment, and may result in closer relationships between different specialists and the development of new skills, knowledge, and learning opportunities while providing some medico-legal protection [3]. Although substantive evidence is still limited, few studies comparing the outcomes of patients treated before and after the implementation of Pituitary MDT (Table 2) highlight some of the benefits [2,36,37].

Grayson and coworkers showed that since the introduction of a Pituitary MDT there has been a reduction in inpatient hospital days, transient diabetes insipidus, a syndrome of inappropriate antidiuretic hormone, hypothyroidism, an unexpected residual tumor, and peri- and post-operative complications [2]. Previously, Carminucci et al. reported that the introduction of Pituitary MDT reduced the length of hospital stay post-operatively from a median of 3 to 2 days without compromising outcomes [37]. Other studies showed that post-operative follow-up by an endocrinologist reduced the risk of 30-day readmission after surgery [38,39]. Cerebrospinal fluid leakage rates after transsphenoidal surgery have decreased since the introduction of a multidisciplinary surgical skull base team in a high-volume Swedish centre [40]. Pituitary MDT benefits have also been highlighted in a series of four women with sellar lesions during pregnancy [41]. Benjamin et al. assessed the cost effectiveness of a post-operative protocol after resection of pituitary tumours implemented by their MDT, concluding that there was a significant cost reduction in laboratory studies (of USD 255.95 per patient); there was also a decrease in the number of patients treated with desmopressin post-operatively after the protocol implementation [36]. Such MDT-based protocols may reduce hospital stays and readmissions, and improve the outcomes and safety of surgical patients [36,37,42,43].

The multidisciplinary management of functioning pituitary tumours, including prolactin, growth hormone (GH)- or adrenocorticotropic hormone (ACTH)-secreting tumours, is crucial for improving clinical outcomes and the prognosis of patients [22,23,44,45,46,47,48]. Experienced neurosurgeons working in Pituitary MDT/PTCOE have better results [49,50,51,52]. Pituitary neurosurgeons achieve higher rates of biochemical cure in acromegaly or Cushing’s disease [51,53,54,55,56,57,58,59]. Moreover, the expertise of neurosurgeons and endocrinologists is crucial in reducing post-operative complications and decreasing the length of hospital stay after operation [33,37], as shown in a surgical series for Cushing’s disease [34].

## 4. Characteristics, Composition and Requirements of the Pituitary MDT/PTCOE

The features, structure, and mode of operation of the MDT vary across the globe and across different conditions [1,2]. Pituitary MDT/PTCOE have only been implemented in recent years; thus, formal definitions are still controversial. However, defining the scope and structure of Pituitary MDT/PTCOE has been addressed by relevant societies in the field, including the Pituitary Society [10,11], and the European Reference Network on Rare Endocrine Conditions (Endo-ERN) [26,27]. 

The endo-ERN subthematic group of hypothalamic and pituitary conditions is based on different MDT and diagnostic and treatment approaches in three key domains: pituitary tumours, congenital hypopituitarism, and acquired hypopituitarism [26,27]. The general characteristics of a Pituitary MDT/PTCOE as defined by the Pituitary Society includes: (i) provide the best care for pituitary tumour patients and pituitary/sellar related disorders; (ii) be independent from health authorities, administrations and for-profit organisations; (iii) be recognised amongst endocrinologists and pituitary surgeons locally, nationally and/or internationally within the endocrine and neurosurgical communities and societies; (iv) advance the science and knowledge in the field of pituitary; (v) provide adequate patient education and community outreach; and (vi) serve as training centre for doctors aiming to specialise in the diagnosis and treatment of pituitary diseases [10].

The general structure of a Pituitary MDT/PTCOE relies on a core team composed of experienced pituitary neurosurgeons and endocrinologists (leading team), supported by specialists in other areas, including neuroradiologists, neuropathologists, neuro-ophthalmologists, otorhinolaryngologists, radiation oncologists, and endocrine nurses [10,60]. More recently, the involvement of neuro-oncologists has also been suggested (Figure 1) [12]. 

### 4.1. Pituitary Neurosurgeons and Neurosurgery Units

Despite recent advances in medical therapy, surgery remains the first choice for pituitary tumours, except for prolactinomas, where dopamine agonists are recommended as primary treatment. Thus, it is unquestionable that the Pituitary MDT/PTCOE depends on an experienced neurosurgeon able to perform endonasal transsphenoidal or transcranial surgical approaches effectively and safely [10,50]. Surgery is the most effective procedure for acromegaly, Cushing’s disease, thyrotrophinomas, resistant prolactinomas, and non-functioning pituitary tumours causing mass effects. It is also recommended for selected cases of apoplexy, Rathke’s cleft cysts, or craniopharyngiomas [10]. 

Being an excellent pituitary neurosurgeon requires solid training and knowledge of the hypothalamic-pituitary and skull base anatomy and physiology, plus continuous practice in a high-volume centre to maintain the expertise. Basic knowledge of neurosurgery relies on a residency program that should cover all neurosurgery-related fields, and not only pituitary [61,62]. Such programs provide limited experience in transsphenoidal pituitary surgery due to the small number of surgeries per year in most centres, as well as insufficient interaction with endocrinologists and other specialists in the Pituitary MDT, which in turn does not allow graduates to obtain enough experience to manage pituitary tumour patients after completing the residency [10,63]. Thus, after the residency, a neurosurgeon seeking to specialise in pituitary should have an additional postgraduate fellowship at an excellent high-volume centre [10,50].

Experienced pituitary neurosurgeons have better outcomes and fewer (and less severe) complications; hence, the number of surgeries performed per year on a continuous basis in a high-volume centre is critical [49,51,52,53,54,55,64,65]. A typical high-volume centre should encounter more than 1000 patients with pituitary diseases per year, with 850 being regarded as the minimal threshold. As defining criteria for a PTCOE, it has been considered 100 pituitary operations per centre per year as the preferred threshold, although 50 surgeries/year are acceptable [11]. Acute post-operative complications, including mortality and readmission rates, should preferably be negligible or nonexistent, with a preferred and acceptable criterion being respectively <2% and <10% of operated patients with complications requiring readmission [11]. However, the reality across European neurosurgical centres is far from such recommendations, with a recent study showing that only 8% of the centres perform more than 100 surgeries per year, whereas more than 40% of centres operate less than 30 patients per year [63]; moreover, most centres operate only 1–5 hormone-secreting tumours per year [66].

The surgeon workload depends on the population size served by the centre and on the number of surgeons existing in the centre. The ideal number of pituitary neurosurgeons per centre has been proposed as 3. However, one surgeon may be acceptable, but this scenario may lead to several issues, such as the centre will be uncovered when that surgeon is absent, the training of fellows may be more difficult, and performing research projects may be impeded or biased [10,11,50]. To avoid these problems while ensuring sufficient workload, the population covered by the centre may be expanded instead of reducing the number of surgeons performing pituitary surgeries; alternatively, neurosurgeons available in a certain region can be concentrated in a single centre [10,11,50]. The preferred population size per PTCOE has been suggested as 3.7 million, while a population of 1.5 million may be acceptable [11]. However, a previous study assessing the optimal numerosity of the referral population defined a PTCOE as a centre where a surgeon would serve a population of at least 9.5 million, which was the size allowing the minimal surgical experience threshold to be reached within 1 years’ time [50].

### 4.2. Endocrinologists and Endocrine Units

Endocrinologists provide high-quality, timely, cost-effective, equitable, accessible, and culturally sensitive health care to patients with endocrine disorders while taking part in the education of patients, families, communities, and authorities to ensure adequate health literacy and decisions/policies [67,68]. Pituitary-focused endocrinologists are crucial to the Pituitary MDT/PTCOE and play a key “holistic” role in the diagnosis, treatment, and follow-up of patients with pituitary disorders. The challenges of pituitary tumours are not only of a surgical nature but encompass a wide range of other issues, such as long-term management of the tumour, treatment-related secondary effects, including hypopituitarism, or diabetes insipidus, and/or morbidity and mortality associated with hormone hypersecretion. Moreover, many pituitary disorders are primarily managed by endocrinologists, such as prolactinomas, congenital hypopituitarism, or acquired hypopituitarism (e.g., after trauma or radiotherapy). Endocrinologists are also crucial in providing support in peri- and post-operative settings, specifically in hormone replacement and water–sodium imbalances [2,10,11,23,39,42,47,69].

Dedicated endocrinologists participating in the Pituitary MDT/PTCOE should have received basic training in internal medicine and endocrinology through the residency, and then performed postgraduate training for at least 12 months in a pituitary tertiary centre. Key skills and competences of a pituitary-focused endocrinologist include: experience in evaluating and managing patients with pituitary diseases, namely those that may impose challenges in the differential diagnosis [46,70,71,72]; thorough knowledge of the laboratory techniques for hormone analysis and capacity to interpret results, including those from dynamic testing; ability to interpret pituitary magnetic resonance imaging (MRI); understand the basics of pituitary pathology [8,73]; awareness of the new developments in the pituitary field in terms of molecular/genetic testing, and in treatment options [28,29]; participation in scientific meetings, and in national or international surveys and registries [27,74,75,76]; willingness to present results at scientific meetings, and advance pituitary science by conducting research, publishing on peer-reviewed journals or textbooks [10,11,42].

The criteria to define the expertise of a pituitary-focused endocrinologist/endocrine unit in an MDT/PTCOE may be more difficult to establish than for pituitary neurosurgeons. However, a recent study encompassing nine PTCOEs highlighted some aspects. The median number of pituitary-focused endocrinologists was 6, ranging from 4 up to a maximum of 17, and the median number of patients managed by the endocrine unit annually was estimated at 1403, varying from 855 up to 1874 patients. These data led to the recommendation of including 6 pituitary-dedicated endocrinologists (4 acceptable) in a Pituitary MDT/PTCOE, overseeing a total of patients that should not be lower than 850 (preferably 1400 patients) [11]. The number of dynamic tests performed by each endocrine unit ranged from 342 up to 4230 (median of 1335) [11]. 

### 4.3. Neuroradiologists and Radiology Units

MRI is the primary imaging modality for the pituitary gland. An experienced neuroradiologist with deep knowledge of the normal anatomy and MRI appearance of the normal pituitary and hypothalamic regions, as well as of pituitary tumours, neoplasms other than pituitary tumours, pituitary tumour-mimicking lesions, and inflammatory/infiltrative stalk processes, is crucial to the Pituitary MDT/PTCOE [77,78]. At least one dedicated neuroradiologist should exist in a Pituitary MDT/PTCOE, although 7 may be preferred [11]. 

The optimal MRI technique relies on thin section T1-weighted sequences in the sagittal and coronal planes before and after gadolinium contrast enhancement; T2-weighted sequences may add useful information in some cases but do not substitute for T1-weighted sequences [79]. A high-field MRI machine with at least 1.5T (or above) and high-resolution should be available at the radiology unit [10,11]. The centre should have access to digital subtraction angiography and bilateral venous sampling of the inferior petrosal sinus, which is critical for many cases of ACTH-dependent Cushing’s syndrome [10,11,46]. The median annual number of pituitary MRI scans and inferior petrosal sinus sampling reported by 9 PTCOE were respectively 810 (range: 125–3411) and 3 (range: 0–20) [11].

### 4.4. Neuropathologists and Pathology Units

Pathology is essential for the diagnosis, management, and follow-up of patients with pituitary tumours and related disorders; thus, one dedicated neuropathologist (preferably three) should be included in the Pituitary MDT/PTCOE [11]. The pathological assessment is crucial to establishing the final diagnosis, and may help guide or determine response to treatment, particularly with the emergence of new molecular biomarkers and targeted therapies [7,28,80,81], as well as in predicting the patient’s prognosis [82,83]. Pathology is key for the differential diagnosis of non-pituitary neoplasms, such as Rathke’s cleft cysts, craniopharyngiomas, pituicytomas, spindle cell oncocytomas, granular cell tumours, meningiomas, or germinomas. Pathology can also establish the diagnosis of infiltrative disorders, such as lymphocytic hypophysitis or Langerhans cell histiocytosis [84].

One or two experienced neuropathologists should be integrated into the Pituitary MDT/PTCOE and provide the final diagnosis for each case using the most updated WHO guidelines [8]. Routine assessment should include information concerning pleomorphism, giant cells, inclusions, inflammatory changes, stroma, hemorrhage, vascular features, granulation patterns, Ki-67, mitotic count, staining for pituitary hormones and transcription factors, and in selected cases, alpha subunit, chromogranin, P53, hormone receptors, or somatostatin receptors [8,10,11].

### 4.5. Neuro-Ophthalmologists and Ophthalmology Units 

Dedicated neuro-ophthalmologists are required for the diagnosis and follow-up of visual impairment in patients with pituitary tumours; thus, it is recommended to include one neuro-ophthalmologist (preferably two) in the Pituitary MDT/PTCOE [11]. A pre-operative evaluation should be offered to patients with visual symptoms or tumours compressing the optic chiasma, and typically include examination of visual acuity, pupil and ocular motility, eye fundus, automated perimetry, and optical coherence tomography [42,85]. 

A complete neuro-ophthalmological examination may help predict the likelihood of visual acuity and visual field deficit improvement, which may occur in 68% and 81% of patients undergoing pituitary tumour resection [86]. Such examination is often useful in establishing the need and urgency of the operation in patients with large tumours, as well as to assess visual outcomes after surgery [42,87,88]. Moreover, a careful examination may identify other causes of visual impairment, such as cataracts, preventing unnecessary pituitary operations or superfluous cataract surgeries [89]. Neuro-ophthalmologists are also essential in the follow-up of pregnant women with macroprolactinomas [90].

### 4.6. Otorhinolaryngologist and Otorhinolaryngology Units

Otorhinolaryngologists are not widely involved in Pituitary MDT/PTCOE, and there is considerable variability regarding their participation among different centres. In 29 of 60 (48.4%) German neurosurgical centres, otorhinolaryngologists are never involved in pituitary surgeries, whereas in only 8 centres (13.3%) surgeries are always performed together with an otorhinolaryngologist [91]. Nevertheless, the collaboration between pituitary neurosurgeons and otorhinolaryngologists during endonasal and other skull base surgical approaches is of extreme value, allowing higher tumour resection rates and fewer post-operative complications, particularly cerebrospinal fluid leakages [39,40,92,93].

Pre-operative assessment by an otorhinolaryngologist is important considering that the surgical approach to the sella is typically performed endonasally (through the nostrils). Hence, anticipating anatomical difficulties or nasal pathologies is relevant for better surgical planning. Moreover, the evaluation of symptoms such as nasal obstruction, rhinorrhoea, or hyposmia, as well as the performance of other tests including nasofibroscopy, nasal function tests, rhinometry, rhinomanometry, and smell tests, may be useful in some cases. Most patients will benefit from endonasal post-operative care, such as crust removal or minor procedures to speed up healing processes and obtain a faster return to a normal quality of life regarding breathing and smelling. Also, these observations should allow early diagnosis of complications such as cerebrospinal fluid leak, sinusitis, reconstruction flap necrosis, or mucocele [42].

### 4.7. Radiation Neuro-Oncologists and Radiotherapy Units

Radiation therapy may be needed to treat pituitary tumour remnants or functioning tumours resistant to medical treatment, as well as other parasellar tumours such as craniopharyngiomas or meningiomas; patients refusing or with contra-indications to surgery may also be eligible for primary irradiation treatment [94,95,96,97,98]. Radiation neuro-oncologists or radiotherapists managing patients with pituitary tumours must have a deep knowledge of the tolerance of the optic system, cranial nerves in the cavernous sinus, temporal lobes, and normal pituitary, and should be available in the Pituitary MDT/PTCOE [10,11]. 

In a survey across 9 PTCOE, it was reported that there was a median number of radiotherapists/radiation oncologists of 3 (range: 2–5), and a relatively low median number of stereotactic radiotherapy and radiosurgery interventions on an annual basis of respectively 5.3 (range: 2–35) and 4.3 (range: 0–60), while conventional radiotherapy was almost fully abandoned [11], likely due to the negative risk:benefit ratio, particularly when compared to newer and more effective techniques [99]. Computer-assisted irradiation techniques should be available at centres of excellence as they remain important tools for selected cases, with rates of local control at 5 years estimated at 94–97% in patients with non-functioning tumour remnants [96], and a likelihood of 50–75% in controlling hormone excess in acromegaly or Cushing’s disease [94,100,101,102].

### 4.8. Other Healthcare Professionals and Units

Recently, the inclusion of neuro-oncologists in the Pituitary MDT/PTCOE has been advocated based on recent molecular and therapeutic advancements, in particular the emergence of new systemic therapies for aggressive or metastatic pituitary tumours such as temozolomide, tyrosine kinase inhibitors, mTOR inhibitors, bevacizumab, or immune checkpoint inhibitors [12]. A median of 3 neuro-oncologists (range: 1–30) have been recently self-reported by 9 PTCOE [11].

Pituitary MDT/PTCOE should include at least one endocrine nurse (preferably 3) [11], considering the importance of nursing care in the pre-, peri-, and post-operative management of pituitary tumour patients, as well as in the long-term follow-up and education of patients with hypopituitarism [10,11,103]. Endocrine nurses are key in facilitating communication between patients and clinicians [104,105]. However, in some pituitary tumour centres of excellence, there are no nurses available [11]. 

There is an emerging role for nuclear medicine in the management of pituitary tumours. Positron emission tomography-computed tomography (PET-CT) with ^18^F-fluorodeoxyglucose may be helpful for differential diagnosis and pre-operative characterization of some sellar lesions. Other tracers are currently used in the pituitary setting [23]. PET-CT with ^68^Ga-labeled somatostatin analogues may localize and determine response to surgical, medical, or radiation treatment, as well as aid in selecting aggressive or metastatic pituitary cases suitable for peptide receptor radionuclide therapy [31]. PET-CT with ^11^carbon-methionine plays an important role in identifying a target not visible (or equivocal) on MRI that might be amenable for surgery or radiotherapy [106,107,108].

Other collaborating medical specialties may be involved in the care of pituitary tumour patients, such as cardiologists, sleep experts, and bone experts, particularly in syndromes related to pituitary hormone excess such as acromegaly or Cushing’s disease [11]. Gynaecologists, and obstetricians might also be crucial in managing cases of pregnant women with pituitary tumours or other sellar lesions [41,109].

## 5. Barriers to the Pituitary MDT/PTCOE

Although the best care model for pituitary tumour patients relies on the Pituitary MDT/PTCOE, its application in the real world often encounters several barriers. Data from a large survey study including 254 neurosurgical centres performing pituitary surgeries across 34 European countries showed that regular pituitary board meetings were held in only 56.3% of them [63]. Hence, more needs to be done to overcome such barriers and to widely implement Pituitary MDT/PTCOE across healthcare systems.

The MDT model is associated with marked consumption of time and resources, and significant costs [3]. In the UK, the estimated cost to the health system of a 2 h monthly Pituitary MDT meeting ranges from £9000 up to £12,000 per year [2]. While such direct costs are easier to calculate based on the time spent and number of physicians involved in such meetings, the indirect financial benefits and cost-effectiveness for high-quality decision making and more efficacious management of pituitary tumours are more difficult to assess, and remain poorly described [2,110]. Recent studies showed that the creation of a Pituitary MDT reduced inpatient hospital days [2,37], decreased post-operative complications [2,40], lowered the risk of readmission after surgery [39], and reduced costs in laboratory studies [36], while at the same time improving clinical outcomes [36,37,42,43]. Multidisciplinary management of patients with functioning tumours achieves higher rates of biochemical cure [51,53,54,55,56,57,58,59], allowing significant savings in terms of their long-term management associated with the costs of medications to treat hormone excess in persistent or recurrent disease. In acromegaly, the costs of somatostatin analogue therapy range between EUR 7900–19,800 per patient depending on dose and country, whereas the annual cost for pegvisomant treatment per patient may vary between EUR 28,300–84,900 in Europe, or be higher than USD 100,000 in the United States [111,112,113]. Also, there is a financial burden associated with the management of pituitary hormone excess-related comorbidities, such as diabetes, hypertension, cardiovascular disease, or mental illness [112,114,115,116,117]. Thus, efficacious and cost-effective care for pituitary tumour patients reduces costs, leading to significant savings aspects often overlooked in the discussions around the Pituitary MDT/PTCOE. 

Efficient communication between the different healthcare professionals involved in an MDT is crucial. The creation of a non-hierarchical and collaborative highly specialised environment, the presentation of the case in a concise and high-quality manner, scheduling structured meetings on a regular basis (once a week, biweekly or monthly, depending on the volume of cases in the centre), and ensuring these meetings are always well-attended by the core members, are key factors for the success of a Pituitary MDT/PTCOE [1,3,10]. The establishment of local guidelines or institutional protocols, and the maintenance of local registries and electronic clinical files are important tools to monitor outcomes and improve when necessary, while also providing a reliable tool to conduct research and communicate scientific results [10].

Another barrier to optimal care for pituitary patients is that health care systems are often too fragmented, lack clinical information capabilities, duplicate services, are not properly designed for chronic care, and may be challenging to modify. In some institutions, the initiatives by endocrinologists or administrators to concentrate procedures on a single neurosurgeon may encounter firm opposition in the Neurosurgery department, situation unlikely to be modifiable given the lack of external guidelines or policies endorsed by authoritative bodies [10]. Moreover, small groups or groups that perform poorly never present or publish their results, thus making it difficult to run audits or independent assessments, which in turn, may also impact scientific advancements [10,27]. There are also demographic factors and disparities related to payer status, insurance, admission types, or geography, imposing barriers to high-quality care that must be overcome to improve the access of patients to centres of excellence [118]. 

## 6. Perspectives of the Different Specialists Involved in our Pituitary MDT

In order to provide additional insights and reflections around this topic from the different medical specialists involved in our Pituitary MDT, we answered two questions: (i) What is the major contribution that your specialty can provide to the Pituitary MDT? (ii) What are the major contributions that other specialties can provide to your clinical practice when managing a pituitary tumour patient? Replies to each question can be found in Appendix A. Overall, the main message emerging from the replies is that this interdisciplinary and collaborative teamwork between the different specialists is crucial for the best decision-making process and delivering high-quality and effective treatment to patients with pituitary diseases. 

## 7. Conclusions

The optimal care for patients with pituitary tumours is best provided in a multidisciplinary and collaborative environment, which requires the contribution of multiple medical specialties working together within the scope of a Pituitary MDT/PTCOE. There are several benefits to providing care to pituitary tumour patients through the pituitary MDT/PTCOE, and this model of delivering care is highly recommended by all relevant societies and international guidelines in the pituitary field. 

Pituitary neurosurgeons and pituitary-focused endocrinologists are normally the leading specialties in the Pituitary MDT and should be supported by other specialties with significant contributions to the diagnosis and management of pituitary diseases, including neuropathology, neuroradiology, neuro-ophthalmology, otorhinolaryngology, and radiation neuro-oncologists, among others. Despite the barriers that may arise when setting up a Pituitary MDT/PTCOE, efforts should be made to overcome them, aiming to deliver high-quality, cost-effective, and safe medical care for all patients with pituitary disorders.

## Figures and Tables

**Figure 1 cancers-16-00950-f001:**
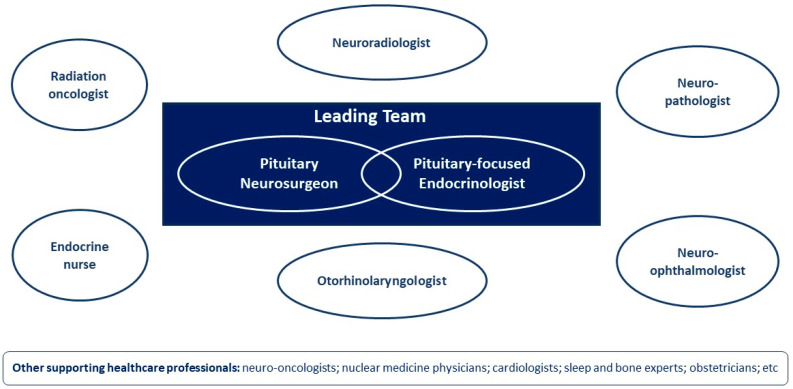
General composition of a Pituitary Multidisciplinary Team (MDT)/Pituitary Tumour Centres of Excellence (PTCOE).

**Table 1 cancers-16-00950-t001:** Mission and main clinical goals of the Pituitary Multidisciplinary Team (MDT)/Pituitary Tumour Centres of Excellence (PTCOE).

**Mission of the Pituitary MDT/PTCOE**	• Providing the best standard of medical care to patients with pituitary diseases • Providing accurate, comprehensive and up-to-date information to patients regarding their conditions • Organising multidisciplinary management, with engagement and collaboration between experienced neurosurgeons and endocrinologists, working together with other supporting medical specialties • Providing education and training to fellows and residents aiming to acquire competences and skills in the management of pituitary diseases • Providing courses, lectures or education initiatives to primary care physicians and other medical specialists, as well as to undergraduate medical students • Compiling data and publishing the results to advance science and knowledge on pituitary diseases • Providing data to regional, national or international registries • Advising health administrators and authorities on problems related to the management of patients with pituitary diseases to improve patient’s experience and safety, and to facilitate care across different healthcare settings
**Main clinical goals of the Pituitary MDT/PTCOE**	• Early detection of the pituitary tumour or pituitary disorder • Establishing the diagnosis and the most suitable treatment for each case, which may be active surveillance, surgery, irradiation and/or medical therapy • For surgical cases, removing the tumour while preserving the normal pituitary tissue and nearby structures, and where appropriate, improving or reverting mass effect symptoms, such as visual defects and/or headache • For patients undergoing surgery, preventing acute complications and readmissions to the hospital • Eliminating or controlling the hormone hypersecretion, preventing its effects in patient’s quality of life and mortality, through surgery alone or in combination with medical treatments and/or radiotherapy • Monitoring and preventing pituitary tumour recurrence • Recognising and managing the acute and delayed complications of the pituitary disease, especially hypopituitarism • Management of complex and potentially life-threatening pituitary conditions, such as pituitary apoplexy, infections, hypopituitarism, or other parasellar pathologies, such as Rathke’s cleft cysts, craniopharyngiomas, chordomas and skull base meningiomas • Remain at the forefront of diagnostic and treatment modalities, including applying the latest developments and technologies in the domains of surgery, molecular, laboratorial, and histopathological testing, radiology, nuclear medicine, radiotherapy, and medical therapy, including the emerging targeted therapeutic options

**Table 2 cancers-16-00950-t002:** Overview of the studies comparing the outcomes/complications of patients with pituitary diseases treated before and after implementing of a Pituitary MDT or multidisciplinary protocols.

Study: First Author, Year, Journal (PMID)	Study Features: Design, Study Population	Main Findings of the Study
Carminucci 2016 *Endocr Pract* (PMID: 26437216) [37]	Retrospective study 214 patients: 113 pre-MDT vs. 101 post-MDT	• Median length of stay in hospital decreased from 3 days pre-MDT to 2 days post-MDT (*p* < 0.01) • Discharge occurred on post-operative day 2 more frequently on the post-MDT group (69 vs. 46%, *p* < 0.01) • Rates of early post-operative DI and readmissions within 30 days for SIADH or other complications did not differ between pre-MDT and post-MDT groups • All patients have received an in-hospital endocrine consultation post-MDT, in contrast with only 40% in the pre-MDT era
Grayson 2021 *J Neurol Surg B Skull Base* (PMID: 34026405) [2]	Retrospective study 279 patients: 89 pre-MDT vs. 190 post-MDT	• Transient DI and SIADH, as well as new secondary hypothyroidism, occurred less often post-MDT (20 vs. 36%, *p* < 0.01; 18 vs. 39%, *p* < 0.01; and 5 vs. 15%, *p* < 0.01, respectively) • Hospital stay was shorter post-MDT (5 vs. 7 days, *p* < 0.001) • Intrasellar residues were less common post-MDT (8 vs. 35%, *p*< 0.001) • Peri- and post-operative complications were more frequent pre-MDT (41 vs. 69%, *p* < 0.001), and were independent of tumour size, hormone status, and surgical technique (OR = 2.14 [1.05–4.32], *p* = 0.04).
Benjamin 2022 *J Neurol Surg B Skull Base* (PMID: 36393880) [36]	Retrospective study 171 patients: 126 pre-protocol vs. 45 post-protocol	• After the implementation of the MDT protocol, there was a reduction in laboratory studies per patient (55.66 vs. 18.82, *p* < 0.001), which corresponded to a cost reduction of USD 255.95 per patient • There was a decrease in the number of patients treated with desmopressin (21.4% pre-protocol vs. 8.9% post-protocol, *p* = 0.04) • All post-protocol patients requiring desmopressin at discharge were identified by 48 h; there was no change in length of stay or need for hydrocortisone post-operatively between the two groups, neither in rates of 30-day readmission
Ghiam 2022 *J Neurol Surg B Skull Base* (PMID: 36393882) [38]	Retrospective study 542 patients: 409 pre-protocol vs. 133 post-protocol	• After the implementation of a MDT post-operative care protocol consisting of post-discharge fluid restriction and close follow-up by an endocrinologist within 1 week of discharge, all-cause readmission decreased (6 vs. 14%, *p* = 0.015); also, patients who were not involved with this MDT post-operative protocol had higher risk of readmission (OR = 2.5 [1.1–5.5]) • Incidence of emergency room visits due to hyponatremia decreased from 3.7 to 0% after implementing the MDT post-operative care protocol (*p* = 0.0279)

DI, diabetes insipidus; MDT, multidisciplinary team; OR, odds ratio; PMID, PubMed identifier; SIADH, syndrome of inappropriate antidiuretic hormone; USD, United States dollar.

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
