# Peer review of "Multidisciplinary Team Care in Pituitary Tumours"

_cancers, 2024, doi:10.3390/cancers16050950_

Round 1

Reviewer 1 Report

Comments and Suggestions for Authors

This manuscript made a thorough revision of the necessity/management of the benefits and advantages of the pituitary multidisciplinary team (MDT) and center of excellence. The text is well written, it is easy to read, and to understand. Some sections are quite long, but this is more a matter of style. As expected, the presence of multiciplinary teams improves the clinical envolution of the patients. However, although is some of the studies it was statistically significant, it was a difference of days. Overall, the presence of MDT is a good thing.

Additional comments:

(1) Regarding "Pituitary tumours arise from the adenohypophysis and account for 10-15% of all intracranial neoplasms". Could  you please expand the histological descrition of pituitary tumors?

You may refer to this link for information:

https://www.pathologyoutlines.com/topic/cnstumorwhoclassification.html

(2) In Section 2. Regarding Pituitary MDT/PTCOE. Are the type of members (specialists) of the multidisciplinary team defined in the guidelines?

(3) Is there any available diagnostic and treatment algorithm for pituitary tumors? Although this manuscript revises more organization aspects, the algorithm may help the readers to understand the complexity of the medical situation.

Author Response

We are grateful to the Reviewer for the positive comments on our manuscript. Worth noting that we reduced some parts of the manuscript, and also we removed tables 3 and 4 from the main manuscript, which overall reduced the length of our revised manuscript. Regarding each one of the additional comments:

  1. We added some considerations about the histology of pituitary tumours, as suggested, referring to the current classification as per tumour cell lineage recommended in the latest WHO 2022 classification.
  2. Although the importance of a pituitary multidisciplinary team is acknowledged in most available guidelines for diagnosing and treating pituitary diseases, there are still few recommendations about their composition and model of function. This results in a wide heterogeneity across different centers and countries. The key papers providing guidance regarding the medical specialists needed in a pituitary MDT/PTCOE are the ones provided by the Pituitary Society, namely in their Position Statement published in 2017, that we cited several times (https://pubmed.ncbi.nlm.nih.gov/28884415; https://pubmed.ncbi.nlm.nih.gov/37640885). In fact, the Pituitary Society identified the key specialists in a Pituitary MDT/PTCOE in their Position Statement in 2017, and suggested later the engagement of neuro-oncologists (https://pubmed.ncbi.nlm.nih.gov/37676531), which are basically the medical specialists we discuss in our paper.
  3. Our manuscript is indeed specifically focused on the organizational aspects of Pituitary MDTs/PTCOE, which we aimed to cover and discuss in detail. However, providing diagnostic and treatment algorithms for pituitary tumours is outside of this paper’s scope and also a challenging task, as each pituitary tumour subtype is diagnosed and managed in a very different way, so we’d have to provide different algorithms for ACTH-secreting pituitary tumours (Cushing’s syndrome), GH-secreting pituitary tumours (acromegaly), prolactin-secreting pituitary tumours (prolactinomas), nonfunctioning pituitary tumours, and TSH-secreting tumours, which would result in a significant expansion of our paper. Moreover, we wouldn’t be able to discuss this in detail, as there are specific guidelines provided by medical societies/associations in the field, such as the Endocrine Society, the Pituitary Society, or the European Society of Endocrinology, which we then referred when appropriated in our manuscript.

Reviewer 2 Report

Comments and Suggestions for Authors

dear colleagues, 

the paper has many shared content and ideation with the Chapter of Athanasios Fountas https://doi.org/10.1016/B978-0-12-819949-7.00050-0

the paper is organized like a book chapter

i suggest that authors run a systematic review of the literature to present novel information

Author Response

Our manuscript consists of a narrative review of the literature, thus certainly will have some common notions and ideation with the mentioned chapter, as it has with other key papers in the field, namely the articles published by the Pituitary Society on PTCOE. We would like to say that, in this particular topic, we deliberately opted to write a narrative review instead of a systematic review or a meta-analysis as it would be difficult to perform these given the lack of original studies on Pituitary MDT/PTCOE. In fact, much of the literature consists of expert opinions or position statements by leading groups or societies, including the Pituitary Society which served as the basis for most content we included in our paper (https://pubmed.ncbi.nlm.nih.gov/28884415; https://pubmed.ncbi.nlm.nih.gov/37640885; https://pubmed.ncbi.nlm.nih.gov/37676531). While we acknowledge that we may not provide a great amount of “novel” information giving the nature of our review manuscript (as any narrative review actually does not), we believe our paper covers this topic in detail, being a comprehensive and informative review gathering all relevant published papers in the domain of Pituitary MDT/PTCOE, including the most recent studies assessing the implementation of Pituitary MDTs or MDT-based protocols (see Table 2), published after the mentioned Book Chapter. Additionally, we believe that our review manuscript will be a useful resource for readers interested in the field, as it gathers all key ideas and concepts around this topic, as well as compiles all key published papers in in our comprehensive references list. However, we acknowledge that we overlooked the mentioned Book Chapter of Athanasis Fountas in our initial reference list, thus for completeness we now added this reference (current reference 13). It also worths noting that the number of review manuscripts published in this topic is very limited, and the literature around this topic is spread across different sources, mainly in specific neurosurgery or endocrine journals focused on pituitary. Thus, our review manuscript being published in Cancers will certainly outreach a wider number of medical specializations and will cover a broader audience, which may be good to deliver key messages on the advantages and benefits of Pituitary MDT/PTCOE, possibly resulting in a better implementation of such multidisciplinary units in many centers around the world. For all the above reasons, we believe that our narrative review paper is relevant, and its publication very timely given the lack of the review manuscripts in this domain, as well as the poor implementation of such Pituitary MDT/PTCOE in many centers or countries, as discussed in detail in our review.

Reviewer 3 Report

Comments and Suggestions for Authors

These authors provide a review of the literature concerning the concepts of the Pituitary MDT/Pituitary Tumour Center of Excellence (PTCOE) in terms of their mission, goals, benefits, structure, proposed models of function and barriers. The authors also provide the views from different specialist doctors involved in their Pituitary MDT. The study is novel. The writing is clear. The literature review is very thorough. The figure is clear and appropriate. The authors provide ample evidence for use of a PTCOE. My only criticism is the use of Tables 3&4. These are very difficult to read. Perhaps the responses could be put in the body of the manuscript. Or, alternatively, the table could include 2 columns, the first being the medical specialist and the second being the question/responses.

Author Response

We are grateful to the Reviewer for the positive comments on our manuscript. We understand the concerns raised regarding the Tables 3 and 4, and we agree these are quite extensive and difficult to read. We thought of adding the tables’ text to the manuscript, but this would result in a very extensive section. Thus, the best solution we found is providing the Tables 3 and 4 as Supplementary Tables, so in our revised manuscript we removed the Tables 3 and 4 from the main manuscript, and we seek to provide these as supplementary tables, converting Table 3 in Supplementary Table 1, and Table 4 in Supplementary Table 2. 

Round 2

Reviewer 2 Report

Comments and Suggestions for Authors

Dear colleagues,

I am pleased with your humbleness and accepting/agreeing that this review offer very little contribution. Thus, your rebuttal letter needs to be rephrased and mentioned in the introduction and limitations.

Since systematic review is not possible at this stage. I suggest that you expand this review on few unexplored areas to justify its publication and avoid redundancy and increase novelty. The areas I can suggest are:

Impact of MDT on cost more explicit in one dedicated section instead of scattered statement.

Impact of MDT on future research.

Impact of MDT on professional development and training. IPE for cancer https://pubmed.ncbi.nlm.nih.gov/30560577/ AND https://pubmed.ncbi.nlm.nih.gov/34723716/

Impact of MDT on burnout/satisfaction.

Discussion need to focus on future guideline establishments and suggestions for studies in areas of gaps.

Author Response

Below, we address the further comments from Reviewer 2 on our revision submission from 3 december 2023, addressing point-by-point each of his/her points.

# Reviewer 2:

Dear colleagues, I am pleased with your humbleness and accepting/agreeing that this review offer very little contribution. Thus, your rebuttal letter needs to be rephrased and mentioned in the introduction and limitations.

We thank the Reviewer 2 for getting back to us so quickly on your recently submitted reply and revised manuscript. We respectfully disagree with the reviewer’s concluding remark saying that our “review offers very little contribution”. We did not say that, and we certainly do not agree with this statement. As previously mentioned, our review manuscript covers this topic of Pituitary MDT/PTCOE in very high detail, being probably the most comprehensive and informative review ever published in this field. It gathers all relevant published papers in the domain of Pituitary MDT/PTCOE, including the most recent studies assessing the implementation of Pituitary MDTs or MDT-based protocols (see Table 2), and we strongly believe that our review manuscript will be a useful resource for readers interested in the field, compiling all relevant ideas and aspects around this topic, as well as all key published papers in in our comprehensive references list. As we also previously mentioned, the number of review manuscripts published in this topic is very limited, and the literature around this topic is spread across different sources, mainly in specific neurosurgery or endocrine journals focused on pituitary. Hence, our review manuscript being published in Cancers will certainly outreach a wider number of medical specializations and will cover a broader audience, which may be good to deliver key messages on the advantages and benefits of Pituitary MDT/PTCOE, possibly resulting in a better implementation of such multidisciplinary units in many centers around the world. Once again, we strongly believe that our narrative review paper is relevant, it will offer an important contribution to the field by summarizing all the literature around this topic and key publications (which is what a narrative review manuscript usually does), and its publication is very timely given the lack of the review manuscripts in this domain, as well as the poor implementation of such Pituitary MDT/PTCOE in many centers or countries, we discussed in detail in our review, and that we hope we manage to successfully raise awareness for. Our opinion of the importance and appropriateness of this review manuscript is actually also supported by the very positive comments and by the high rating scores from Reviewer 1 and Reviewer 3, which seem to indeed align with our own positive perspective on our paper, and which unfortunately seem to contrast with the reluctancies of Reviewer 2 regarding our manuscript.

Hence, for all these reasons we respectfully don’t agree with the need of rephrasing our rebuttal letter, neither to specifically mention any of these aspects or limitations to our review manuscript.

Since systematic review is not possible at this stage. I suggest that you expand this review on few unexplored areas to justify its publication and avoid redundancy and increase novelty. The areas I can suggest are:

Impact of MDT on cost more explicit in one dedicated section instead of scattered statement.

Impact of MDT on future research.

Impact of MDT on professional development and training. IPE for cancer https://pubmed.ncbi.nlm.nih.gov/30560577/ AND https://pubmed.ncbi.nlm.nih.gov/34723716/

Impact of MDT on burnout/satisfaction.

We thank the Reviewer 2 for the comment and for the suggestions made. However, we respectfully disagree with the suggestion of expanding further our manuscript, as we feel that our review is already quite extensive and wrote in high detail, which includes about 5000 words already and more than 100 references. In fact, in this revised version we attempted to slightly reduce the text instead, because some sections were quite long (as also noted by Reviewer 1). Additionally, in a narrative review it might not be appropriate to navigate through unexplored areas or areas not studied at all (i.e., where original data is missing) because you may deviate from the purpose of reviewing the literature and starting making new claims and originating new data/hypothesis/theories (without being supported by real evidence), which may indeed violate the fundaments of a manuscript of this nature. Of note, we have some detail considerations about costs of Pituitary MDT and its potential impact on the financial/economic system of health care settings (as much as the current literature tells us), we also emphasized the importance of MDT in research, for example by noting the need of academic productivity (papers, presentations, etc) or for instance by noting that more than one neurosurgeon is needed to perform unbiased research projects, etc, and we also noted the role of Pituitary MDT in the formation and training of professionals; additionally, we said that Pituitary MDT is also important for professional fulfilment for example. Hence, we feel that we touched upon all the items brought up by the Reviewer 2, however we cannot do any further deep analysis or considerations, and especially certainly not calculating the impact of the Pituitary MDT as there are no specific original studies on these topics in Pituitary MDT to indeed measure such impacts and to allow robust and evidence-based conclusions.

Hence, for all the above-mentioned reasons, we respectfully disagree with the Reviewer 2 about expanding any further our manuscript, and we hope that its current revised version we submitted 2 days ago are of his/hers please, and he/she can see some of the merits we tried to highlight in the above-mentioned text.

Round 3

Reviewer 2 Report

Comments and Suggestions for Authors

no more comments.